# An Optical Sensory System for Assessment of Residual Cancer Burden in Breast Cancer Patients Undergoing Neoadjuvant Chemotherapy

**DOI:** 10.3390/s23125761

**Published:** 2023-06-20

**Authors:** Shadi Momtahen, Maryam Momtahen, Ramani Ramaseshan, Farid Golnaraghi

**Affiliations:** 1School of Mechatronic Systems Engineering, Simon Fraser University, Surrey, BC V3T 0A3, Canada; shadi_momtahen@sfu.ca (S.M.); maryam_momtahen@sfu.ca (M.M.); ramasesh@sfu.ca (R.R.); 2Department of Medical Physics, BC Cancer, Abbotsford, BC V2S 0C2, Canada

**Keywords:** biosensor, machine learning, breast cancer, chemotherapy, residual cancer burden

## Abstract

Breast cancer patients undergoing neoadjuvant chemotherapy (NAC) require precise and accurate evaluation of treatment response. Residual cancer burden (RCB) is a prognostic tool widely used to estimate survival outcomes in breast cancer. In this study, we introduced a machine-learning-based optical biosensor called the Opti-scan probe to assess residual cancer burden in breast cancer patients undergoing NAC. The Opti-scan probe data were acquired from 15 patients (mean age: 61.8 years) before and after each cycle of NAC. Using regression analysis with k-fold cross-validation, we calculated the optical properties of healthy and unhealthy breast tissues. The ML predictive model was trained on the optical parameter values and breast cancer imaging features obtained from the Opti-scan probe data to calculate RCB values. The results show that the ML model achieved a high accuracy of 0.98 in predicting RCB number/class based on the changes in optical properties measured by the Opti-scan probe. These findings suggest that our ML-based Opti-scan probe has considerable potential as a valuable tool for the assessment of breast cancer response after NAC and to guide treatment decisions. Therefore, it could be a promising, non-invasive, and accurate method for monitoring breast cancer patient’s response to NAC.

## 1. Introduction

Breast cancer is a common disease that causes significant morbidity and mortality worldwide, with an estimated 43,250 women expected to have died from breast cancer in the US in 2022 [1]. Neoadjuvant chemotherapy (NAC) is a standard treatment for locally advanced and inoperable breast cancer [2]. However, measuring tumor response to NAC is challenging, and no gold standard exists for assessing tumor response before surgery [3]. Pathological complete response (pCR) is often used as a marker of chemosensitivity and survival outcomes [4,5]. Still, it does not provide precise information on the amount of residual tumor [6,7], making it difficult to predict overall survival [8] accurately.

The residual cancer burden (RCB) method is a pathological approach that quantifies the residual disease in the breast or regional lymph nodes after chemotherapy [9]. RCB provides a standard for classification of the extent of residual disease based on a computed RCB index that uses the tumor bed area (TBA), the extent of in situ disease (IS) and invasive cancer (IC), and the number of involved lymph nodes (LNs) [9]. RCB is scored on a scale of 0–3, with RCB-0 (RCB score = 0) corresponding to pCR, RCB-I (0 < RCB score ≤ 1.36), RCB-II (1.36 < RCB score ≤ 3.28), and RCB-III (RCB score > 3.28) [10]. However, RCB has been highly validated as reproducible and prognostic, small cohorts limit its accurate prognosis in breast cancer subtypes [7].

Near-infrared (NIR) biosensing devices, such as diffuse optical tomography (DOT), have shown promise for breast cancer detection and monitoring. These techniques utilize the unique optical properties of biological tissue to detect biomarkers and molecular changes associated with cancer lesions. NIR-DOT is a non-invasive, easy-to-use point-of-care bedside tool [10] that can potentially measure responses to NAC for clinical decision-making and prognostication [11], providing critical indicators of treatment response, such as tumor hemoglobin distributions [12] and metabolism [13]. For example, researchers used NIR-DOT with ultrasound localization to image heterogeneous hemoglobin distribution in large breast cancers [14]. In contrast, others introduced a compact ultrasound-guided NIR-DOT system for non-invasive imaging of breast cancer [15] and used both MRI and diffuse optical tomography to improve breast cancer detection [13]. However, accurately predicting optical parameters from conventional NIR-DOT imaging techniques can be challenging due to nonlinear photon scattering and fewer known measurements than unknowns [16].

To address this challenge, ongoing research is being conducted at Simon Fraser University (SFU) to develop a proprietary near-infrared optical breast-scanning (Opti-Scan (US Patent: US20220409058—HANDHELD PROBE AND SYSTEM FOR IMAGING HUMAN TISSUE)) probe to detect breast cancer. The Opti-Scan probe collects backscattering light intensity from the breast surface to identify abnormalities [10]. Four generations of the probe have been designed and successfully tested for cross-sectional imaging. The first generation introduced in 2014 utilized eLEDs with a multiwavelength pointed-beam illumination source, and subsequent developments led to the 2016 generation, which underwent clinical trials on patients with known breast cancer, demonstrating high accuracy at the Jim Pattison Outpatient Care and Surgery Centre (JPOCSC) [10]. The current Opti-Scan probe was developed to further enhance the probe’s performance, incorporating improvements in the linear CCD sensor, light intensity control, noise reduction, and data collection process. In addition, the imaging system of the previous version, which was based on the diffusion equation (DE) and had some limitations, was improved by introducing the modified diffusion equation (MDE). This improved imaging system accurately determines absorption values and provides clearer images of various breast phantoms [17] and breast tissues [18]. However, since the MDE was slow in real-time imaging, a machine learning (ML) algorithm was developed to predict scattering coefficients of breast tissues [19], and an ML ensemble method was used to classify liquid phantoms into healthy and unhealthy classes [20]. The absorption coefficient needs to be measured to enable image reconstruction and accurate calculation of RCB. Therefore, in collaboration with BC Cancer (BCCA) in Abbotsford, we developed a new method that utilizes a machine learning algorithm to estimate the absorption coefficients, potentially improving the accuracy of the Opti-Scan probe. This method was tested on fifteen breast cancer patients undergoing neoadjuvant chemotherapy (NAC) and monitored periodically by the probe. The ML model accurately differentiated between cancerous and non-cancerous tissues, demonstrating the probe’s potential as a non-invasive diagnostic biosensor for breast cancer detection.

The RCB index/class is predicted according to calculations of the area under the curve of residual disease based on changes in each patient’s optical properties. This approach shows promise in accurately classifying patients based on their response to treatment and predicting their prognosis. Utilizing the Opti-Scan probe with the RCB index/class prediction methods shows promise as a more comprehensive and non-invasive method for monitoring breast cancer treatment efficacy and improving patient outcomes. However, further studies with larger patient cohorts are needed to validate these findings and determine the clinical utility of this approach.

## 2. Materials and Methods

### 2.1. NIR Opti-Scan Probe for Breast Tissue Imaging and Cancer Detection

The NIR Opti-Scan probe (Figure 1) is a portable device designed for optical imaging of breast tissues. It is a non-invasive imaging modality that utilizes the near-infrared (NIR) region of the electromagnetic spectrum, where biological tissues’ absorption and scattering properties are relatively low, allowing for deeper tissue penetration and imaging [17]. The probe consists of a charge-coupled device (CCD) with 2048 active pixels and a 14 µm pixel pitch, providing an effective imaging area of 28.672 mm (2048 × 14 µm). Two encapsulated light-emitting diodes (eLEDs) are mounted 15 mm away from the two sides of the CCD detector to provide symmetrical light illumination. Each eLED can illuminate light with selectable wavelengths of 690, 750, 800, and 850 nm in the NIR region [21]. These specific wavelengths were selected based on the large difference between extinction coefficients in the four primary components in breast tissue, including oxyhemoglobin (HbO2), deoxyhemoglobin (Hb), fat, and water (H_2_O) [22]. Previous studies have shown that the absorption coefficients of these components depend on the wavelength [12,23]. Thus, the selected wavelengths allow for optimal sensitivity to the breast tissue composition. For more details about the probe specifications, please refer to Table 1.

During imaging, reflectance versus the distance to the light source is collected by the CCD and acquired by a computer simultaneously. The software interface communicates with the probe via USB 2.0 and contains a graphical user interface (GUI) for the operator’s ease of use. The GUI comprises a series of tabs and data graphs for various operations, including manual and periodic light source selection, CCD integration time setting, and data collection mode [21].

The image reconstruction algorithm is a crucial NIR handheld Opti-Scan probe component. The algorithm involves the collection of raw data obtained from 2048 pixels in the CCD array, which are acquired from the surface of the tissue. These raw data are then processed using a series of computational steps to reconstruct an image of the tissue’s internal optical properties. To reduce noise in the raw data, the reflectance data are averaged from 16 contiguous pixels, resulting in 128 points of raw reflectance data. Then, the tissue’s optical properties, including absorption and scattering coefficients, are determined by fitting the experimental reflectance obtained by the CCD to the theoretical reflectance values calculated by the modified diffusion equation (MDE) solution. Finally, the optical parameters calculated for each position in the tissue are converted to cross-sectional optical images.

The Opti-Scan probe’s potential for non-invasive and accurate imaging of biological tissues, particularly for breast cancer diagnosis and monitoring, has been demonstrated in various studies. One such study presented a reconstructed 2D image based on the absorption parameters for a physical phantom with a spherical abnormality of size 4.5 mm in the center, as shown in Figure 2 [17]. In another recent study [18], the Opti-Scan probe’s MDE imaging was used to measure the optical properties of 15 breast cancer patients accurately. The probe used different slices of images to map each patient’s blood concentrations to optical 3D imaging reconcentration, as shown in Figure 3a. The 3D optical images of a patient were captured using 12 slices at 690 nm, and a 3D volume model of the tumor was created using MATLAB rendering capabilities, as shown in Figure 3b. These findings highlight the Opti-Scan probe’s potential for precise breast cancer detection and monitoring.

### 2.2. Study Desing and Scanning Procedure

This preliminary study included fifteen female patients (mean age: 61.8 years) diagnosed with early breast cancer and eligible for neoadjuvant intravenous systemic therapy at BC Cancer Abbotsford. The patients received six to eight cycles of treatments, and optical scans were performed separately for each patient before and after each treatment cycle to evaluate the tumor’s response to treatment. Two trained Ph.D. students conducted the optical scans, following a standardized scanning procedure established by the scanning team. The procedure included the determination of the reference location, the optimal direction of the scan, the probe’s location and orientation, the number of slices required to capture the whole tumor, and the interval distance between each slice (see Figure 4). The optical scans were performed on both the cancerous lesions and the healthy regions of the contralateral breast, with the reference selected based on the approximate location of the tumor underneath the skin.

To evaluate the accuracy of the optical measurements, three metrics were compared: pathology, palpation (PALP), and ultrasound (US). Pathological information about each tumor was obtained through a needle biopsy, which served as the gold standard for tumor size measurement. The oncologist assessed the tumor size and texture by palpation during each visit to determine the treatment progress. The patients received ultrasound screening before, during, and after the treatment cycle to evaluate the tumor’s response to treatment. Table 2 presents the tumor size (cm) before and after treatment measured with palpation and ultrasound for a subset of 3 of 15 enrolled patients. As an example, the ultrasound measurement of patient 13′s tumor size during the pretreatment screening was 5.0 × 5.1 × 4.1 cm (length × width × depth).

### 2.3. Machine Learning Models for Breast Optical Properties and Residual Cancer Burden

In this study, we aimed to use an innovative approach to measure pathological residual cancer burden (pRCB) values, considering the complexity of pRCB measurements. We utilized the Opti-Scan probe to measure optical RCB values, referring to them as optical residual cancer burden (oRCB). By estimating breast tissue’s optical properties, including absorption and scattering coefficients, we established the correlation between pRCB and oRCB values. The absorption coefficients were measured using reflectance data obtained from the probe, serving as the basis for determining the optical RCB value and predicting corresponding pathological RCB values. To accurately calculate optical properties based on reflectance measurements, were utilized machine learning (ML) techniques.

#### Review of Current ML Methods for Breast Optical Properties and Residual Cancer Burden

This section reviews several ML models used to estimate breast optical properties and residual cancer burden. Conventional image reconstruction algorithms estimate optical tissue property distributions ($\;{\hat{x}}^{*}$) by minimizing the regularized objective function in Equation (1) [24]:(1)$${\hat{x}}^{*}=argmin\;\left\{\frac{1}{2}{\left|\left|F\left(\hat{x}\right)-y\right|\right|}_{2}^{2}+\lambda R\left(\hat{x}\right)\right\}\;$$
where λ is the regularization hyperparameter, and R (·) is a regularization term, usually the standard Tikhonov regularization ($R\left(x\right)=\lambda {\left|\left|x\right|\right|}^{2}$). These model-based algorithms are computationally expensive, limiting their practicality and real-time applications [25].

Recently, machine learning applications have shown the potential to improve the accuracy of reconstructed images and solve inverse scattering problems [26]. For example, Feng et al. [27] proposed a multilayer perceptron (MLP) feedforward neural network for 2D-DOT image reconstruction, but its performance was reduced significantly for limited-angle acquisition. Yoo et al. [28] designed a convolutional neural network for 3D-DOT inverse scattering problems to determine the nonlinearity of the inverse scattering problem. Machine learning algorithms have also been used to estimate residual cancer burden (RCB). Ref. [29] highlights the potential of machine learning with multiparametric MRI (mpMRI) to predict the complete pathological response (pCR) to neoadjuvant chemotherapy (NAC) in breast cancer patients. In another study [30], deep-learning-derived volumes of locally advanced breast cancer on MRI showed comparable performance to functional tumor volume in predicting residual disease after chemotherapy (AUC = 0.76). These findings suggest the potential of deep-learning-based segmentation for accurate assessment of tumor load and residual cancer burden in breast cancer patients. A related study [31] investigated artificial intelligence-based segmentation of residual tumor burden after neoadjuvant therapy as an objective and reproducible solution for tumor response scoring. Despite various ML models for residual breast measurement using MRI and other technologies, there remains a need to evaluate breast optical properties and residual cancer burden, specifically through diffuse optical tomography (DOT) technologies. To address this need, our study was focused on leveraging highly accurate ML algorithms and optimizing the performance of the Opti-Scan probe for accurate estimation of breast optical properties and residual cancer burden.

### 2.4. Proposed ML Model for Breast Optical Properties and Residual Cancer Burden

In this study, we aimed to predict optical properties in breast tissue using regression analysis. The training dataset (D) consisted of 640 phantom data collected using various breast phantoms, each with 128 different reflectance, scattering, and absorption values. These phantoms simulate breast tissue with varying absorption levels, which can mimic the optical properties of normal and cancerous breast tissue. The absorption coefficients were measured using the MDE [17], which utilizes the modified diffusion equation to measure the optical properties of phantoms.

The test dataset $\left(T\right)$ contains 15 sets of testing data, each comprising 128 data reflectance values collected by the probe for each patient, denoted as $X_{1},X_{2},\ldots ,X_{128}$. The ML-optical model aims to train the training dataset based on phantom data and predict the corresponding unknown absorption coefficients ($Y_{1},Y_{2},\ldots ,Y_{128}$) of normal and cancerous breast tissue based on the reflectance values obtained from the breast phantoms.

To estimate the optical properties, we employed a predictive model (F) that uses k-fold cross-validation, where the training set was split into k smaller subsets. Given the exponential nature of the reflectance curve in the patient data, polynomial regression was selected as the appropriate method. To determine the optimal polynomial degree, we fit polynomials of varying degrees to the training data and compared their performance on the testing data.

We employed the gradient descent algorithm during the training process and specified two important hyperparameters: the learning rate and the iteration count. The learning rate controlled the step size for each iteration, and we set it to 0.5 in this study. The iteration count determined the number of times the model updated the weights and biases, and we performed 10,000 iterations to ensure convergence and optimize the model’s performance. We monitored the convergence by calculating the cost function at each iteration. We visualized the results by plotting the cost against the number of iterations, demonstrating the successful training of the polynomial regression model with a degree of 4.

To assess the accuracy of our model, we evaluated two metrics on the testing data: the mean square error (MSE) and the $R^{2}$ score. The MSE was calculated to quantify the average squared difference between the predicted and actual absorption coefficients, yielding an impressive value of 0.011%. Additionally, the $R^{2}$ score, which measures the proportion of the variance in the target variable explained by the model, demonstrated an accuracy of 90%. These metrics provided valuable insights into the model’s performance in capturing the optical coefficients for normal and cancerous breasts across patients, slices, and treatments using eLED1 and eLED2.

Figure 5 depicts the path of light through breast tissue; two light paths from eLED1 and eLED2 converge at $\left(X1,Y1\right)$ to measure the absorption coefficient as a superposition of the coefficients of both eLED. Figure 6a displays absorption curves for healthy and unhealthy breast tissue by combining estimated absorption properties obtained by each eLED. The oRCB values were subsequently derived by subtracting absorption values from cancerous breast tissue from corresponding normal tissue values for all patients, slices, and treatments. The area under the curve (AUC) of residual disease was computed from the absorption difference (i.e., error) between healthy and unhealthy tissue values for all slices in each treatment, as illustrated in Figure 6b.

This dataset and methodology formed the foundation for developing and evaluating the machine learning (ML) model. The ML model uses reflectance data to predict the optical properties of breast tissue, which, in turn, allows for the estimation of optical residual cancer burden. The oRCB values are correlated with the pRCB index. To calculate the oRCB values for fifteen patients, we compared the AUC values of residual disease for each treatment. Then, we divided the smallest value (usually for the last treatment) by the largest value (usually for pretreatment).

Furthermore, another regression analysis was employed to determine the unknown pRCB values based on available values. The new training dataset consisted of eight oRCB values and their eight corresponding known RCB values obtained from the BC Cancer Agency. This training dataset was used to develop a regression model. The test dataset, which we used to evaluate the model’s performance, consisted of seven oRCB values as input and seven corresponding unknown RCB values as outputs. We could predict the unknown pRCB values for each of the seven cases by applying the regression model to the test data.

## 3. Results: Machine-Learning-Based Method for Breast Tissue Optical Property Determination and Treatment Response Monitoring

### 3.1. Optical Property Determination

In this section, we present our study’s results in using a machine learning (ML) predictive model to determine the optical properties of healthy and unhealthy breast tissues. Figure 7 shows the absorption curves for each patient’s healthy and unhealthy breasts, represented by the black and red curves. The black curves were used as a baseline to detect changes in optical parameters. The red curve should fit the black curve when there is no tumor. However, deviations indicate changes in the optical properties due to the tumor’s presence.

#### 3.1.1. Patient 12

For patient 12, the ML model indicated that slice #4 had the highest absorption concentrations, consistent with palpation (palpation reported slice #4 as the reference). The noticeable changes in optical properties observed in slices #3 to #5 are clear evidence of the presence of the tumor.

#### 3.1.2. Patient 13

For patient 13, the model indicated that slice #7 (or #8) reflected the highest absorption concentrations. The largest optical changes reflecting the tumor were observed in slices #5 to #9.

#### 3.1.3. Patient 26

For patient 26, the absorption curve in slice 1 was large, likely due to the nipple effect. This patient had two tumors, and palpation reported slice #3 as the reference. The model found a tumor on the right side of slice #3, representing a higher concentration of the larger tumor, but the smaller tumor was not reflected in this slice. Slice #6 showed the largest total changes in optical properties since it showed both tumors’ optical properties.

### 3.2. Treatment Response Monitoring and Residual Cancer Burden (RCB)

In this section, we present the results of the ML model for treatment response monitoring and prediction of residual cancer burden in breast tissues. Figure 8 shows the AUC of residual disease for all slices and treatments for selected patients, providing valuable information about patient response to treatments.

#### 3.2.1. Patient 12

As shown in Table 2, the tumor volume decreased and became not-palpable (NP) after the second post-treatment scan, indicating a complete response (pRCB# = 0). The ML model confirmed the reduction in tumor size after the second post-treatment scan and further validated the continued reduction, indicating a complete treatment response (pRCB# = 0).

#### 3.2.2. Patient 13

Patient 13 received seven cycles of NAC, and the tumor volume decreased gradually (pRCB# = 0), also confirmed by the ML model. The model indicates that patient 13 responded completely to the chemotherapy drugs without any residual cancer burden (pRCB# = 0).

#### 3.2.3. Patient 26

Patient 26 had multiple tumors in her right breast, some shrank during chemotherapy, but others remained after the treatment. Despite undergoing six treatment cycles, residual tumors were observed in post-treatment scans. As shown in Table 2, the tumor volume decreased initially on the post-chemotherapy scan, but the subsequent scans showed incomplete elimination of the tumor. The ML model confirmed the presence of residual disease after neoadjuvant chemotherapy.

Table 3 shows the estimated AUC of residual disease for the healthy and unhealthy sides of three patients across all slices and treatments and their corresponding errors. The oRCB values were calculated as the proportion of the area containing cancer by comparing the AUC values of residual disease for each treatment and dividing the smallest value (usually for the last treatment) by the largest value (usually for pretreatment).

Table 4 displays the known pRCB values and classes, with “NA” representing the unknown values we aimed to calculate using Opti-Scan. The information provided in Table 3, including the errors for treatments, is also shown in Table 4 for all patients, allowing us to estimate the oRCB for all 15 patients. Negative oRCB values were replaced with zero since they are not biologically plausible. Then, the unknown pRCB values for fifteen patients were predicted using AUC of residual disease values and oRCB values, with patient 18 excluded due to the lack of data. Table 4 also indicates the predicted pRCB values and the corresponding classes using cut points at 0, 1.36, and 3.28.

For model training, the dataset consisted of known oRCB values as features and corresponding known pRCB values as targets. Regression analysis was used to predict the unknown pRCB values based on the corresponding known oRCB values.

Table 5 compares the oRCB and the predicted pRCB values for the selected patients, considering the initial unknown pRCB and predicted values. The table also provides the corresponding lower and upper confidence intervals, indicating the certainty range associated with the predicted pRCB values.

Figure 9 presents the prediction of pRCB values using oRCB values. Panel (a) shows the estimated oRCB values for patients with known pRCB values, while panel (b) presents the predicted pRCB values for patients with an unknown pRCB index. The scatter plot displays the relationship between the oRCB and pRCB values. The blue dots represent the oRCB values, while the red dots represent the predicted pRCB values. The solid red line represents the linear regression model’s predictions. The light gray area indicates the confidence intervals for the predicted pRCB values, providing a range of certainty associated with the predictions. The model achieved an accuracy of 98.301%, as measured by the coefficient of determination ($R^{2}$), indicating a strong correlation between the oRCB and predicted pRCB values. Additionally, the model’s mean squared error (MSE) is 2.775%, representing the average squared difference between the actual and predicted pRCB values.

The oRCB values for all fifteen patients were consistent with the corresponding pRCB values, indicating the effectiveness of the ML model in predicting treatment response. The absorption differences (i.e., total errors) between healthy and unhealthy tissue values for all slices in each treatment concentration change inside a tumor provided valuable information about patient response to treatments. These results demonstrate the potential of ML-based methods to predict residual cancer burden and monitor treatment response in breast tissue.

## 4. Discussion

The results of this study indicate that the ML model is a reliable and effective tool for the determination of the optical properties of breast tissue. Our findings suggest that the ML-generated optical properties can be used to obtain optical images, which may aid in diagnosing breast cancer. However, further studies are needed to investigate the clinical utility of our ML model for breast cancer diagnosis.

In terms of treatment response monitoring, the ML method showed potential in detecting gradual tumor volume reduction and confirming complete treatment response in patients 12 and 13. However, limitations were observed in patient 26, for whom the ML method was not able to detect remaining tumors after treatment completion. Combining the ML and MDE [17] methods may enable more accurate treatment response monitoring in breast tissue.

Furthermore, the predicted pRCB values were found to be highly accurate, with an accuracy of 0.98%, which is an important clinical implication of this study. The ML model can provide a more objective and quantitative assessment of treatment response, allowing for earlier identification of non-responders and potential adjustments to the treatment plan.

Although the findings of this study are promising, it is important to note that the training dataset size was relatively small and limited to breast phantoms, with only fifteen patients tested. Therefore, further studies with larger dataset sizes and diverse patient populations are needed to validate the findings. Additionally, while we used a regression analysis algorithm in this study, evaluating the performance of other ML algorithms, such as transfer learning and ensemble learning, in this context would be beneficial.

Overall, the results of this study suggest that ML has the potential to improve the accuracy and objectivity of breast cancer treatment response monitoring and RCB estimation. However, further studies are needed to confirm these findings and evaluate the clinical utility of the ML model for breast cancer diagnosis and treatment.

## 5. Conclusions

In conclusion, using the Opti-scan probe, this study demonstrates the successful application of machine learning methods to assess residual cancer burden (RCB) in breast cancer patients undergoing neoadjuvant chemotherapy (NAC). The use of regression analysis and cross-validation allowed for the calculation of the optical properties of healthy and unhealthy breast tissues. The ML model trained on the optical parameter values and breast cancer imaging features obtained from the Opti-scan probe data achieved a high accuracy of 0.98 in predicting RCB number/class based on changes in optical properties. These findings suggest that the Opti-scan probe has considerable potential as a valuable non-invasive tool for monitoring breast cancer patients’ response to NAC and guiding treatment decisions. In future work, we plan to design machine-learning models based on patients’ optical data to recreate optical blood concentration images. Overall, this study provides a strong foundation for further investigation into using machine-learning-based NIR biosensors for assessing residual cancer burden in breast cancer patients.

## Figures and Tables

**Figure 1 sensors-23-05761-f001:**
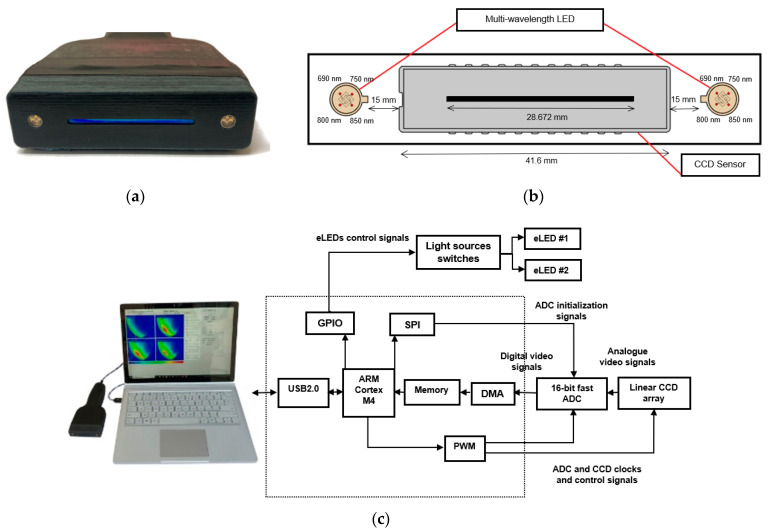
Overview of the Opti-Scan probe and system: (**a**) front-end view of the probe, showing the light sources and linear CCD; (**b**) diagram illustrating the CCD sensor and LED arrangement in the head of the probe; (**c**) photograph of the Opti-Scan probe connected to a laptop running custom software and a schematic diagram of the sensory system, highlighting the different components.

**Figure 2 sensors-23-05761-f002:**
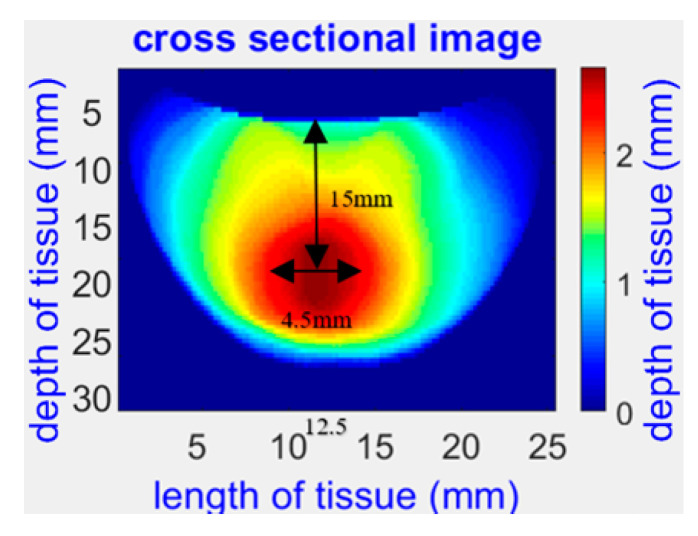
Optical image of a physical phantom with a 4.5 mm spherical abnormality at the center captured at 690 nm [17].

**Figure 3 sensors-23-05761-f003:**
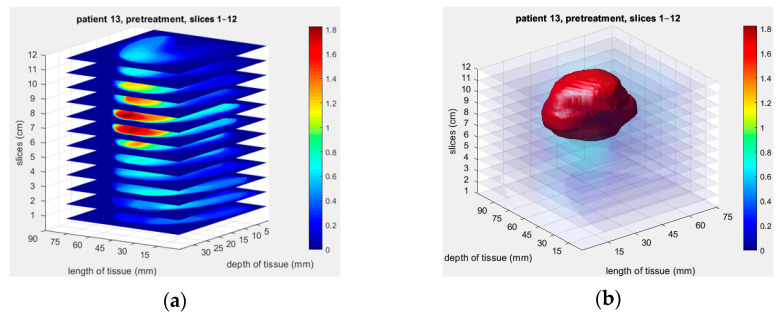
Reconstructed 3D optical image of a patient’s tumor using 12 slices at 690 nm: (**a**) reconstructed image using the Opti-Scan probe’s MDE imaging; (**b**) 3D volume model of the tumor created using MATLAB rendering capabilities, with gaps between the adjacent slices interpolated [18].

**Figure 4 sensors-23-05761-f004:**
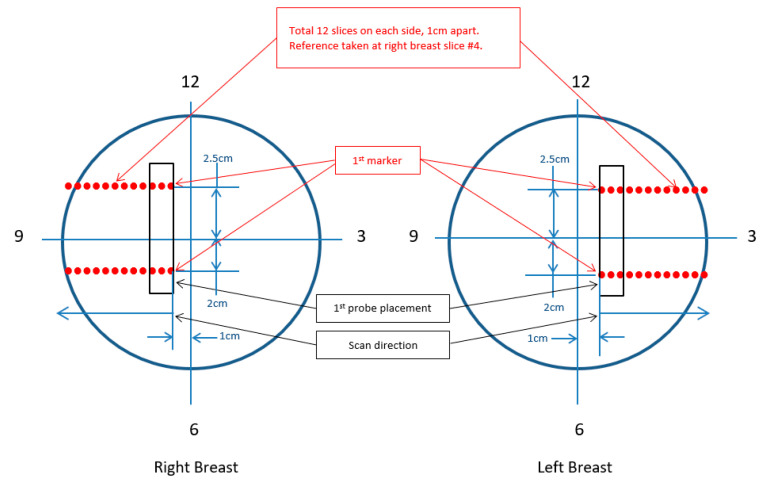
Tumor localization scanning procedure. This figure shows the tumor localization scanning, including the determination of the reference location, optimal scan direction, probe placement, and number of slices needed.

**Figure 5 sensors-23-05761-f005:**
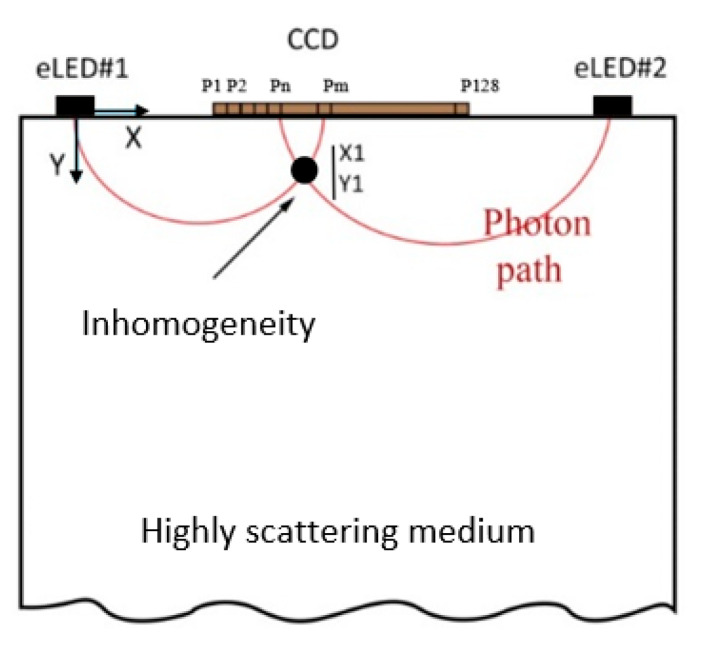
Sketch of light propagation and absorption measurement in breast tissue using two eLEDs [17].

**Figure 6 sensors-23-05761-f006:**
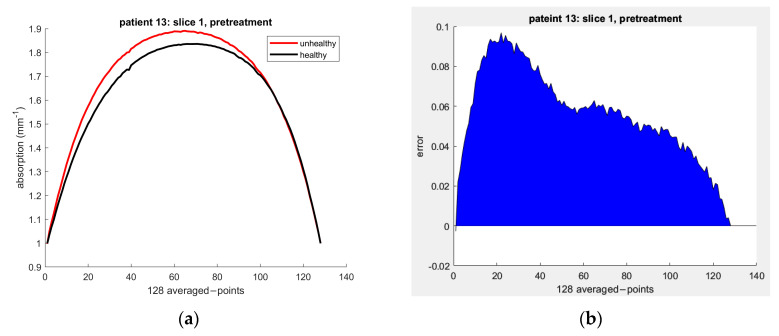
Absorption curves of patient 13′s healthy and unhealthy breast tissue for slice 1 during the pretreatment screening at 690 nm: (**a**) the absorption curves of healthy (black) and unhealthy (red) tissue; (**b**) the AUC for residual disease calculated by subtracting the absorption values of the cancerous breast from the corresponding values of normal tissue.

**Figure 7 sensors-23-05761-f007:**
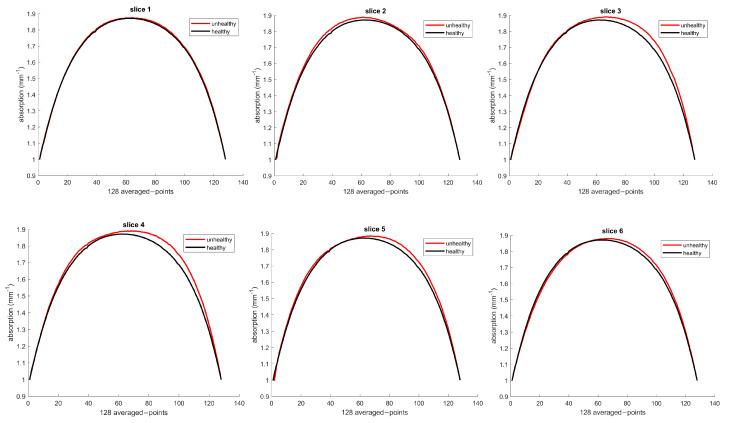
Absorption curves at 690 nm were generated by ML for patients (**a**) 12, (**b**) 13, and (**c**) 26 during pretreatment screening. Black curves: healthy breast tissue. Red curves: unhealthy tissue affected by the tumor. Deviations from the black curve signal change in optical properties due to tumor presence.

**Figure 8 sensors-23-05761-f008:**
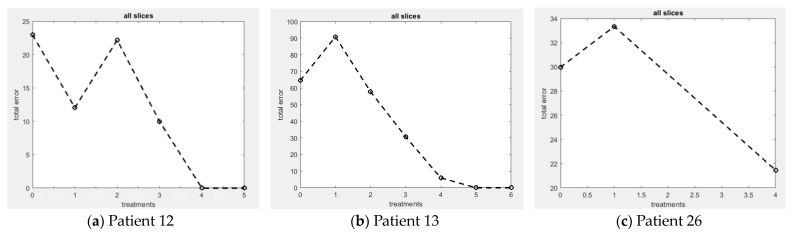
ML-based treatment response monitoring for all slices and treatments at a wavelength of 690 nm in patients (**a**) 12, (**b**) 13, and (**c**) 26, showing the absorption differences (total errors) between healthy and unhealthy tissue values and providing valuable information about patient’s response to treatments.

**Figure 9 sensors-23-05761-f009:**
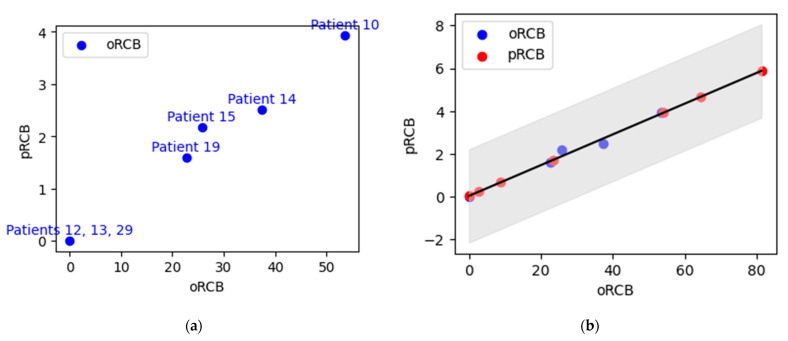
Prediction of unknown pathological RCB values using optical RCB values: (**a**) the estimated oRCB values for patients with known pRCB values; (**b**) the pRCB values predicted for patients with unknown pRCB values. The light gray area indicates the confidence intervals for the predicted pRCB values.

**Table 1 sensors-23-05761-t001:** Specifications of the NIR handheld Opti-Scan probe.

Parameter	Value
Imaging technology	Near-infrared optical imaging
Detector type	Linear CCD
Detector resolution	2048 pixels
Image resolution	128 × 128 pixels
Imaging area	28,672 mm (2048 × 14 µm)
Pixel pitch	14 µm
Detector sensitivity	1800 (V/Lx.S) @ 660 nm
Light source	Encapsulated light-emitting diodes (eLEDs)
Wavelengths	2 × (690, 750, 800, and 850 nm)
Distance from CCD	15 mm
Max. frame rate	24
Power consumption	100 mA @ 5V
Radiated power	20 mA

**Table 2 sensors-23-05761-t002:** Patient characteristics and treatment progress: tumor size (cm) before and after treatment measured with palpation (PALP) and ultrasound (US). Note: “NP” indicates “not palpable” and “NA” indicates “not available.

Patient	Tool	Tumor Size (cm)							
		Pretreatment	Post- Treatment 1	Post- Treatment 2	Post- Treatment 3	Post- Treatment 4	Post- Treatment 5	Post- Treatment 6	Post-Chemo
12	PALP	2.5 × 2.5	3 × 3	NP	NP	NP	NP	NP	NA
	US	3.2 × 1.3 × 2.0	NA	NA	NA	NA	NA	NA	NA
13	PALP	10 × 9	7 × 8	5 × 6	3.5 × 3.5	2.5 × 2	2.5 × 2	NP	NA
	US	5.0 × 5.1 × 4.1	NA	NA	NA	NA	3.2 × 1.4 × 1.7	NA	NA
26	PALP	8 × 10 3	4 × 3	NA	5 × 5.5	3	NA	5.5 × 5.5	NA
	US	3.8 × 3.9 × 2.3 1.9 × 1.7 × 1.9	NA	NA	NA	1.0 × 1.3 × 0.9 1.0 × 1.0 × 0.5	NA	NA	0.9 × 1.0 × 0.7 0.4 × 0.4 × 0.5

**Table 3 sensors-23-05761-t003:** Estimated AUC of residual disease for healthy and unhealthy sides of patients 12, 13, and 26 for all slices and treatments, along with corresponding errors.

Patient	AOC	Pretreatment	Post- Treatment 1	Post- Treatment 2	Post- Treatment 3	Post- Treatment 4	Post- Treatment 5	Post- Treatment 6
12	Unhealthy	1680.1	1669.2	1679.3	1667.2	1649.3	1652.1	NA
	Healthy	1657.1	1657.1	1657.1	1657.1	1657.1	1657.1	NA
	Error	22.96	12.04	22.17	9.98	0.00	0.00	NA
13	Unhealthy	1902.9	1929.2	1896.1	1869.0	1844.2	1833.2	1821.5
	Healthy	1838.4	1838.4	1838.4	1838.4	1838.4	1838.4	1838.4
	Error	64.56	90.88	57.77	30.63	5.89	0.00	0.00
26	Unhealthy	1633.6	1627.3	NA	NA	NA	NA	1627.2
	Healthy	1603.7	1593.0	NA	NA	NA	NA	1605.7
	Error	39.97	34.34	NA	NA	NA	NA	21.46

**Table 4 sensors-23-05761-t004:** Prediction of unknown pathological residual cancer burden (pRCB) values and classes using optical residual cancer burden (oRCB).

Patient	Known pRCB Value	Known pRCB Class	Treatment	Unhealthy	Healthy	Error	Predicted oRCB	Predicted Unknown pRCB Value	Predicted Unknown pRCB Class
10	3.93	RCB-III	Pre-t	1914.8	1839.3	75.45	53.54	NA	NA
			Post-t7	1882.9	1842.4	40.40			
12	0.00	RCB-0	Pre-t	1680.1	1657.1	22.96	0.00	NA	NA
			Post-t5	1652.1	1657.1	0.00			
13	0.00	RCB-0	Pre-t	1902.9	1838.4	64.56	0.00	NA	NA
			Post-t6	1821.5	1838.4	0			
14	2.51	RCB-II	Pre-t	1657.7	1614.1	43.64	37.37	NA	NA
			Post-t7	1630.4	1614.1	16.31			
15	2.18	RCB-II	Pre-t	1707.1	1601.6	105.51	25.75	NA	NA
			Post-t7	1630	1602.8	27.176			
16	NA	NA	Pre-t	2144.2	1983.3	160.95	81.34	5.88	RCB-III
			Post-t3	2122.5	1991.6	130.92			
17	NA	NA	Post-t1	2035	1981.5	53.463	8.75	0.67	RCB-I
			Post-t7	1995.6	1990.9	4.679			
18		Lack of Data							
19	1.6	RCB-II	Pre-t	1860.5	1815.4	45.15	22.7	NA	NA
			Post-t5	1825.6	1815.4	10.25			
21	NA	NA	Post-t4	1603.5	1588.8	14.69	54.15	3.93	RCB-III
			Post-t7	1615.5	1607.6	7.96			
22	NA	NA	Post-t1	2034.5	1995.8	38.72	23.47	1.73	RCB-II
			Post-t6	2014.5	2005.4	9.09			
26	NA	NA	Post-t1	1627.3	1594.0	33.34	64.37	4.66	RCB-III
			Post-t6	1627.2	1605.7	21.46			
29	0.00	RCB-0	Post-t2	2074.2	2073.3	0.89	0.00	NA	NA
			PC	2028	2054.8	0.00			
30	NA	NA	Pre-t	2018.2	1983.3	34.91	2.67	0.23	RCB-I
			Post-t5	1976.6	1975.7	0.93			
31	NA	NA	Pre-t	2025.3	1999.3	25.99	0.00	0.00	RCB-0
			Post-t5	2001.1	2015.3	0.00			

**Table 5 sensors-23-05761-t005:** Comparison of oRCB and predicted pRCB values with their corresponding lower and upper confidence intervals for selected patients.

Patient	oRCB Value	pRCB Value	Lower Confidence Interval	Upper Confidence Interval
16	81.34	5.88	3.70	8.06
17	8.75	0.67	−1.51	2.85
21	54.15	3.93	1.75	6.11
22	23.47	1.73	−0.45	3.91
26	64.37	4.66	2.49	6.84
30	2.67	0.23	−1.94	2.41
31	0.00	0.00	−2.13	2.22

## Data Availability

Not applicable.

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
