# Peer review of "An Optical Sensory System for Assessment of Residual Cancer Burden in Breast Cancer Patients Undergoing Neoadjuvant Chemotherapy"

_sensors, 2023, doi:10.3390/s23125761_

Round 1
Reviewer 1 Report
The study focused on the development of a machine learning-based method for determining the optical properties of healthy and unhealthy breast tissue. The authors used absorption curves to detect changes in optical parameters, which could indicate the presence of a tumor. The model was able to accurately predict the optical properties of breast tissue for three patients, and the results were further validated by comparing them to known clinical outcomes, such as the response to treatment and residual cancer burden.
The study's limitation is the relatively small sample size, as mentioned by the authors, which may restrict the generalizability of the findings. Nevertheless, the results presented indicate a promising potential for machine learning-based approaches in predicting the optical characteristics of breast tissue and monitoring treatment response.
However, while the result provides a clear line of reasoning and conclusion, there are a few things that could be added to provide a more complete picture:
- The results section could benefit from further clarification in certain areas. it can be inferred from the context that there may be some confusion or uncertainty about the relationship between the DOB cancer burden percentage and the predicted RCB values.The authors used a method called regression analysis to predict unknown RCB values based on existing RCB values and DOB-generated cancer burden values. The testing dataset was used to assess the accuracy of their method. Although the authors explain their approach in detail, it is unclear how the DOB cancer burden percentage is connected to the RCB values in LINE 230-240 & Table 4.
- Table 3 could benefit from additional data visualization to better understand the relationship between the AUC values and the different treatments, such as color coding or highlighting could help to highlight patterns and trends.
- Table 4 could be improved by including additional information such as confidence intervals or p-values to indicate the level of certainty in the predicted RCB values. Additionally, the table could benefit from visual aids such as color coding or highlighting to make it easier to read and interpret the data.
- In subsection 2.3, the authors could also provide a more detailed explanation of the ML-DOB model and its components. For example, the authors could describe the architecture of the neural network used in the model and provide more information on the hyperparameters used during training.
Reviewer 2 Report
This manuscript entitled “An Optical Sensory system for Assessing Residual Cancer Burden in Breast Cancer Patients Undergoing Neoadjuvant Chemotherapy” presented an approach by the application of machine learning methods to assess residual cancer burden (RCB) in breast cancer patients undergoing neoadjuvant chemotherapy (NAC). This preliminary study was included fifteen female patients (mean age, 61.8 years) diagnosed with early breast cancer and eligible for neoadjuvant intravenous systemic therapy at the BC Cancer Abbotsford. The use of regression analysis and cross-validation allowed for the calculation of the optical properties of healthy and unhealthy breast tissues. The ML model used in the manuscript is trained on the optical parameter values and breast cancer imaging features obtained from the DOB-scan probe data, achieved a high accuracy of 0.98 in predicting RCB numbers/classes based on changes in optical properties. These findings suggesting the DOB-scan probe has great potential as a valuable non-invasive tool for monitoring breast cancer patient's response to NAC and guiding treatment decisions. This study would be interested and can be published in the present form.
Reviewer 3 Report
This review accompanies “An Optical Sensory system for Assessing Residual Cancer Burden in Breast Cancer Patients Undergoing Neoadjuvant Chemotherapy” by Momtahen, et al. The writing in this paper is clear and methodical. The article presents the results of a study that used a custom optical tomography device and machine learning model to look at breast cancer patients. In particular, the authors highlight data that tracks 3 patients through their chemotherapy treatment.
While this is a strong article, it would benefit from addressing the following concerns:
1. In Table 1, specifications that are an improvement from the previous generation should be clearly indicated. In the introduction, lines 59-60, you state that there were previous generations of this probe. It is unclear to the reader what these improvements specifically are.
2. In Table 2, abbreviations of PALP and US are not provided.
3. It would help the reader to have the training dataset and rationale behind the size and design described in a bit more detail. For example, why do the authors believe 640 training datasets is sufficient? How were these datasets created? Are these datasets permutations of other acquired images using the same device? Using a different device? Or, from a different study? Or were they acquired on a physical phantom? An image of a physical test phantom (if used) would be helpful.
4. In lines 248-251, the authors describe how to interpret the absorption curves for the patients. It would be helpful to know if a higher absorption than the baseline indicates something slightly different than when the peaks of the polynomial are out of phase (as with patient 13, pretreatment slice 2 vs patient 13, pretreatment slice 9).
5. It may be clearer to report Error as percentage in Table 4.
6. In lines 350-353, it would be great to compare the spatial resolution of the imaging technique to the size of the remaining tumors (if the approximate size of these tumors is known).
7. In lines 362-364, the authors discuss comparing other techniques. It would be helpful to include some citations in which other ML algorithms were used in a similar context as support for this reasoning and to add more context to the discussion.
This paper is well written. I have no English grammar/language concerns.
Round 2
Reviewer 3 Report
This manuscript has been improved. The context for the work is more clear and the findings are better described. It is a strong article.
The writing is clear.